

# Interleukin 35 induced Th2 and Tregs bias under normal conditions in mice

Xiaoning Zhang[1,*], Zhiqiang Zhang[2,*], Zhiqiang He[1], Mingyan Ju[1], Jiaci Li[1], Jinghua Yuan[1], Yaqing Jing[1], Keqiu Li[1], Yi Liu[1] and Guang Li[1]

[1] Department of Genetics, School of Basic Medical Sciences, Tianjin Medical University, Tianjin, China
[2] Department of Pathology, Tianjin Hospital of ITCWM, Nankai Hospital, Tianjin, China
[*] These authors contributed equally to this work.

## ABSTRACT

**Objective**. The benefits of IL-35 treatment have been verified in multiple animal models of diseases, while its influence on T cells immunity under normal condition still needs to be elucidated. The present study was designed to investigate the effects modulating IL-35 levels *in vivo* and *in vitro* on T cells, response and also the effects on T cells subsets in normal mice.

**Methods**. A plasmid pMSCV-IL-35-GFP carrying mouse linear IL-35 fragment with two subunits joint together was constructed and the heterodimer expression was confirmed. Normal mice were randomly divided into three groups and received an intravenous injection of PBS, pMSCV-GFP and pMSCV-IL-35-GFP respectively. After 72 h, spleen tissues and peripheral blood were harvested for following analysis. Meanwhile, splenic T cells were isolated and incubated with 10, 30, or 50 ng/mL recombinant IL-35 factor for 24 h with the addition of anti-CD3/CD28 *in vitro*. T-cell subsets were assessed by Fluorescence activated cell sorting (FACS) and related cytokines together with effector molecules were determined by real time PCR.

**Results**. Western blotting confirmed a 52 kDa band in the cell lysate of HEK 293T transducted with pMSCV-IL-35-GFP plasmid, indicating a successful expression of IL-35. Ebi3 and IL-12A, two subunits of IL-35, could be identified 72 h post DNA injection. IL-35 upregulation *in vivo* effectively inhibit CD4$^+$ and CD8$^+$ T cell proliferation and Th1 cytokine secretion. Effector molecules of CD8$^+$ T cells were also remarkably suppressed. On the contrary, high level of IL-35 significantly induced CD4$^+$ CD25$^+$ Tregs and Th2 enhancement. The *in vitro* study provided similar results.

**Conclusion**. The results indicated Th1 and CD8$^+$ T cell inhibition and Th2 and Tregs bias in the presence of IL-35 under a normal state which partly contributed to its therapeutic potential.

Corresponding authors
Yi Liu, xiaoyurly@163.com
Guang Li, lig@tmu.edu.cn

## INTRODUCTION

Interleukin 35 (IL-35), a heterodimer composed of Epstein-Barr-virus-induced gene 3 (Ebi3) and interleukin-12 alpha (IL-12A), secreted by natural regulatory T cells (Tregs), is a novel cytokine of the IL-12 family (*Collison et al., 2007*; *Sawant, Hamilton & Vignali, 2015*). Unlike the pro-inflammatory properties of other IL-12 family members, IL-35 plays potent immunosuppressive roles partly by means of Tregs expansion, which is
essential for the maintenance of immune tolerance (*Collison et al., 2010*). Recent studies have demonstrated the efficacy of IL-35 in inflammatory bowel disease (*Wang et al., 2018*), autoimmune encephalomyelitis (*Choi et al., 2017*; *Guan et al., 2017*), collagen-induced arthritis (*Li et al., 2016*) and acute graft-versus-host disease (*Zhang et al., 2015*). Our team also proved the therapeutic application of IL-35 in dextran sulfate sodium (DSS)-induced colitis (*Zhang et al., 2018*). Thus, the anti-inflammatory ability makes IL-35 a promising intervention agent in inflammation, infection and other immune-related disorders.

T cells especially $CD4^+$ T helper (Th) cells and $CD4^+$ $CD25^+$ Tregs are critical for immune-regulation in cellular immunity and immunity homeostasis (*Xiao et al., 2012*; *McQuillan, Lynch & Mills, 2010*). Cytokine profiles released by these cells also take part in and accelerate their action (*Choi et al., 2015*; *Egwuagu et al., 2015*). Th1 cells together with secreted IL-2 and interferon-$\gamma$ (IFN-$\gamma$) have been well-known to mediate inflammation and related diseases, which got significantly inhibited by IL-35 in animal disease models (*Ma et al., 2014*; *Guo et al., 2017*). Th2 and Tregs, in favor of anti-inflammation partly depending on the production of IL-10 and IL-35, respectively, received effective *increase* post IL-35 administration (*Guo et al., 2017*; *Zhao et al., 2017*; *Bettini & Vignali, 2009*). Another $CD8^+$ T cells, whose function relies on cytotoxin release including Granzyme B (Gzmb) and perforin 1 (Prf1) upon antigen stimulation, also received effective suppression following IL-35 use (*Wong & Pamer, 2003*; *Milstein et al., 2011*). However, most of these findings about the immune-regulation capacity of IL-35 were discovered based on disease modeling animals, the *in vivo* effects on normal animals have not been well described. In the present study, a plasmid carrying recombinant mouse IL-35 sequence was intravenously injected into normal mice and the short-term effect of general overexpression of IL-35 heterodimer on immunological status, particularly the differentiation of T-cell subsets, was evaluated. The results could facilitate illuminating the underlying therapeutic mechanisms of IL-35.

## MATERIALS AND METHODS

### Reagents

RPMI 1640 and fetal bovine serum (FBS) were purchased from Biological Industries (Cromwell, CT, USA). Phosphate buffered saline (PBS) and erythrocyte lysis buffer were bought from Beijing Solarbio Science & Technology Co., Ltd. (Beijing, China). FuGENE transfection reagent was acquired from Promega Biosciences (Sunnyvale, CA, USA). IL-12 p35 antibody came from Abcam (UK). TRIzol was from Thermo Fisher Scientific Inc. (Waltham, MA, USA). Phanta Super-Fidelity DNA Polymerase, HiScript II Q Select RT SuperMix for qPCR (+gDNA wiper) and AceQ qPCR SYBR Green Master Mix were purchased from Vazyme Biotech Co., Ltd. (Nanjing City, China). Anti-CD3 and CD28 for T cell stimulation were bought from Affinity Biosciences (Cincinnati, OH, USA). Anti-mouse FITC-CD4, PE-CD8 and PE-CD25 for flow cytometry were acquired from Thermo Fisher Scientific Inc. (USA). The plasmids pMSCV-GFP and pcDNA3.1-IL-35 were cryopreserved in our laboratory. Recombinant human IL-35 (rIL-35) was from PeproTech (Rocky Hill, NJ, USA).

## Animals

Wide type male C57BL/6J mice (8- to 12-weeks old, Certificate SCK 2014-0013) were purchased from the Academy of Military Medical Sciences (Beijing, China). Mice were kept under specific pathogen free conditions according to institutional guidelines. Experimental protocols and animal care methods were subjected to approval by Animal Care and Use Committee of Tianjin Medical University (TMUaMEC 2017012).

## Cell cultures and treatment *in vitro*

Lymphocytes were collected from the spleen tissues of wild type C57BL/6 mice. In brief, naive T cells were cultured in RPMI 1640 without FBS, supplemented with 100 mg/mL streptomycin, 100 U/mL penicillin and two mM L-glutamine in a 37 °C incubator with 5% $CO_2$ post erythrocyte lysis buffer treatment. To investigate the effect of IL-35 on T-cell subsets differentiation *in vitro*, recombinant human IL-35 (10, 30 or 50 ng/mL) was added to the cell culture medium appended with anti-CD3 (3 μg/mL)/CD28 (2 μg/mL). Untreated cells served as a control. After 48 h, cells were harvested for real time PCR detection.

## Construction of the plasmid carrying IL-35 gene

The mouse IL-35 sequence (1.4 kb) covering *Ebi3* linked with *Il12a* was a kind gift from Prof. Jiyu Ju (Weifang Medical University, Shandong). The full IL-35 coding gene was amplified by PCR and subcloned into a vector (pMSCV-IRES-GFP) by *EcoRI/XhoI* double digestion to construct IL-35 expression plasmid pMSCV-IL-35-GFP. The fragment was confirmed by DNA sequencing. SDS-PAGE and western blot for IL-12 p35 detection in HEK293T cells were used to confirm the recombinant IL-35 expression 48 h post transfected with the plasmid.

## Administration of pMSCV-IL-35-GFP

Mice were randomly divided into three groups with similar mean body weight: PBS group, pMSCV group and pMSCV-IL-35 group (five mice per group). Briefly, mice received an intravenous injection of 300 μL PBS containing 50 μg pMSCV-GFP, 50 μg pMSCV-IL-35-GFP or nothing. After 72 h, the spleen tissues were harvested and the tissue homogenates were lysed by erythrocyte lysis buffer to remove the red blood cells. The splenic T-cell products served for real time PCR and flow cytometry assay.

## Real time PCR

Total RNA of the spleen tissues and splenic T cells was extracted using Trizol reagent and two μg RNA was reverse-transcribed using HiScript II Q Select RT SuperMix for qPCR (Vazyme Biotech, China) according to the manufacturers' instructions. To evaluate the expression of involved genes, quantitative real time PCR detection was undertaken with AceQ qPCR SYBR Green Master Mix. All the specific primers were synthesized from Sangon Biotech Co., Ltd. (Shanghai, China), and the sequences of each primer were listed in Table 1. Each sample was executed and analyzed in triplicate. GAPDH was used as the endogenous control.

**Table 1  Primer sequences for real time PCR detection.**

| Genes | Sense primers (5′→3′) | Antisense primers (5′→3′) | Product length (bp) |
|---|---|---|---|
| Ebi3 | GTT CTC CAC GGT GCC CTA C | CGG CTT GAT GAT TCG CTC | 100 |
| Il12a | CCA CCC TTG CCC TCC TAA A | GCC GTC TTC ACC ATG TCA TCT | 121 |
| Il2 | CGG CAT GTT CTG GAT TTG AC | TCA TCA TCG AAT TGG CAC TC | 134 |
| Ifng | CTG ATC CTT TGG ACC CTC TG | ACA GCC ATG AGG AAG AGC TG | 121 |
| Il10 | GCC TTA TCG GAA ATG ATC CA | TGA GGG TCT TCA GCT TCT CAC | 115 |
| Gzmb | GAC CCA GCA AGT CAT CCC TA | CCA GCC ACA TAG CAC ACA TC | 186 |
| Prf1 | CGG TGT CGT GTG GAA CAA TA | TCA TCA TCC CAG CCG TAG TC | 126 |
| Foxp3 | CTG CCT TGG TAC ATT CGT GA | CCA GAT GTT GTG GGT GAG TG | 101 |

Notes.

Ifng, interferon $\gamma$; Gzmb, granzyme b; Prf1, perforin; Foxp3, forkhead box P3.

## Fluorescence activated cell sorting (FACS) analysis

FACS examination, proceeded as described previously (*Sun et al., 2015*; *Chen et al., 2016*), was used to determine the subpopulations of CD4$^+$, CD8$^+$ T cells and CD4$^+$CD25$^+$ Treg cells in cultured T cells and splenic T cells post IL-35 treatment. In simple terms, cell suspensions under analysis were incubated with anti-mouse fluorescein isothiocyanate (FITC)-CD4, -fluorescein phycoerythrin (PE)-CD8 and -CD25 antibodies respectively at 4 °C for 30 min. The stained-positive cells were assayed using flow cytometer (BD FACS Calibur; San Jose, CA, USA). FlowJo 7.6.1 software was utilized for followed data analysis.

## Statistical analysis

Data was presented as the mean ± standard error of the mean (Mean ± SEM). And statistical analysis was proceeded using SPSS 17.0 software (SPSS Inc., Chicago, IL, USA). One-way ANOVA was used in comparison among groups and a post-hoc contrasts by Student–Newman–Keuls test was applied to confirm the significance. Two-tailed $P$ values < 0.05 were considered statistically significant.

## RESULTS

### The constructed plasmid gained successful gene expression of IL-35

As shown in Fig. 1A, a recombinant plasmid carrying IL-35 gene composted of *Ebi3* and *IL12a* fragments was constructed. The expression of IL-35 was further confirmed by HEK293T transient transfection assay. SDS-PAGE and western blot for IL-12 p35 were conducted and the results both showed strong bands at about 52 KDa in DNA transfected cell lysates (Fig. 1B), indicating the successful expression of IL-35.

### IL-35 inhibited CD4$^+$ T cells and promoted CD4$^+$ CD25$^+$ Tregs generation *in vitro*

It has been reported that IL-35 could act on phenotype differentiation of CD4$^+$ T cells (*Guan et al., 2017*; *Huang et al., 2017*). In the present study, splenic T cells were isolated and directly treated with 10, 30, or 50 ng/mL recombinant IL-35 factor for 24 h in the presence of anti-CD3/CD28 *ex vivo*. The cells were then collected to assess the proportion

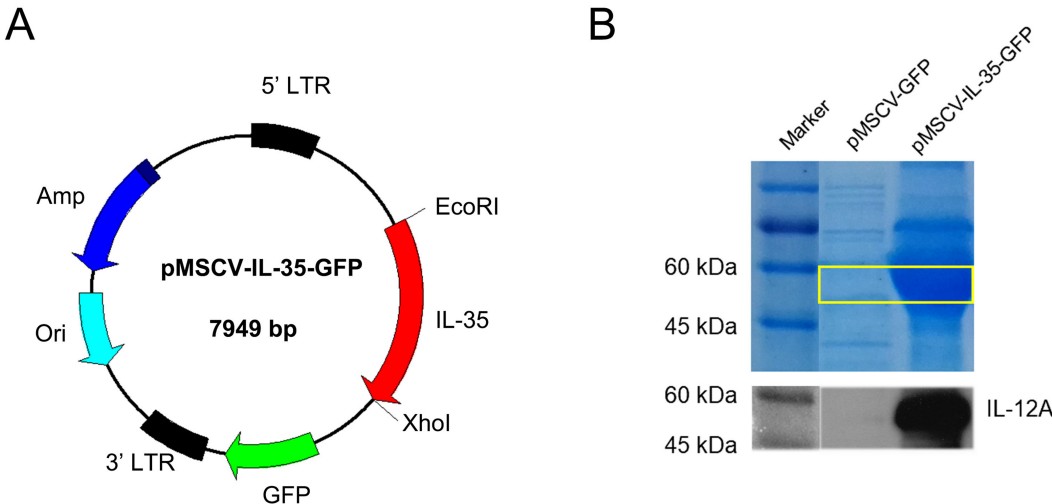

**Figure 1   Characterization of mouse IL-35 expression.** (A) Construction schematic of pMSCV-IL-35-GFP (7,949 bp) encoding the recombinant IL-35. (B) IL-35 expression (52 kDa) was confirmed by SDS-PAGE and western blotting 48 h post transfection into HEK 293T cells.

of T-cell subsets by FACS. The results showed that CD4$^+$ T cell population substantially raised upon activation, which got effectively reduced post rIL-35 supplement (Fig. 2A). Also, the CD4$^+$CD25$^+$ Tregs proportion decreased when exposed to anti-CD3/CD28, but increased after recombinant IL-35 treatment (Fig. 2B). In addition, the rIL-35 affect showed some dose-dependence, indicating that IL-35 could inhibit CD4$^+$ T cells proliferation and generate CD4$^+$CD25$^+$ Tregs.

## IL-35 facilitated Th2 and Tregs function *in vitro*

Cytokines produced by Th and Treg cells also take part in and play important roles in immune-regulation (*Ma et al., 2014*). The total RNA of cultured T cells mentioned above were extracted and the cytokine expression involved Th1 (*Il2*), Th2 (*Il10*) and Tregs (*Foxp3*) were examined by real time PCR. Data indicated that rIL-35 administration significantly reduced *Il2* level and sharply boosted *Il10* and *Foxp3* expression in a dose-dependent manner (Fig. 3), which suggested that IL-35 treatment facilitated Th2 and Tregs function.

## IL-35 overexpression inhibited CD4$^+$ and CD8$^+$ T cells but enhanced CD4$^+$CD25$^+$ Tregs *in vivo*

To further investigate the effect of IL-35 on the T cells differentiation *in vivo*, wild type C57BL/6J mice received an intravenous injection of the plasmid pMSCV-IL-35-GFP. After 72 h, peripheral blood was sampled to test the expression of *Ebi3* and *Il12a* fragments by real time PCR. Spleen tissues were used to estimate the proportion of T-cell subsets by FACS. The results demonstrated both significant increases in *Ebi3* and *Il12a* levels in mice received pMSCV-IL-35-GFP injection, suggesting the expression of exogenous gene sequence and upregulation of IL-35 (Fig. 4). The same as Fig. 2 described, FACS analysis showed obviously lowered percentage of total CD4$^+$ and CD8$^+$ T cells (Figs. 5A–5D)
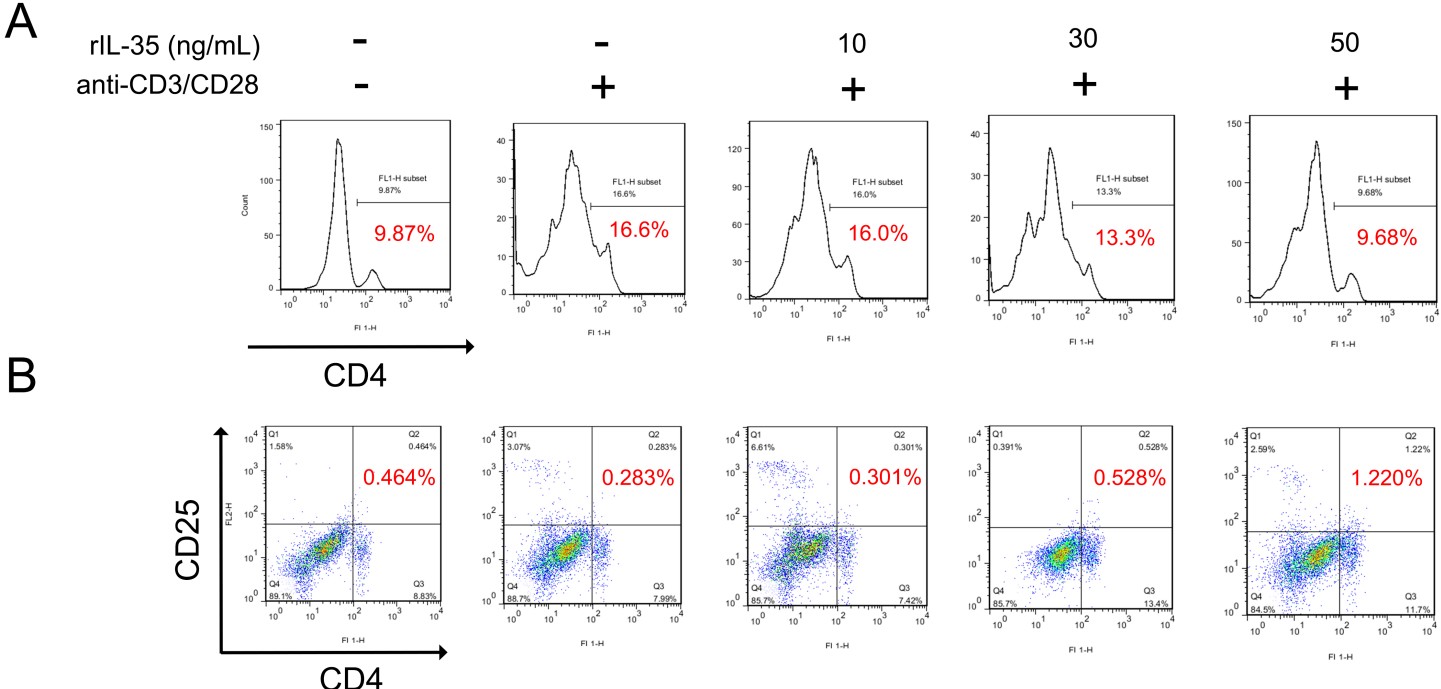

**Figure 2   The phenotype of total CD4⁺ T cells and CD4⁺ CD25⁺ Tregs detected by FACS.** (A) CD4⁺ T cells proliferation got effective inhibition with rIL-35 addition. (B) IL-35 promoted the amplification of CD4⁺ CD25⁺ Tregs. Both the effect on proliferation of CD4⁺ T cells and CD4⁺ CD25⁺ Tregs appeared in some dose-dependent manner.

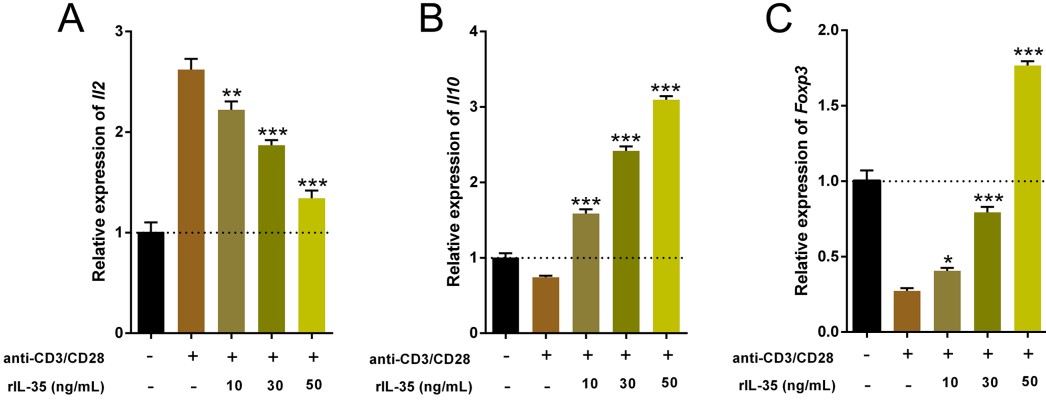

**Figure 3   The expression of *Il2*, *Il10*, and *Foxp3* in splenic T cells following rIL-35 addition for 24 h *in vitro*.** Remarkably lower *Il2* level (A) but multiplied *Il10* (B) and *Foxp3* (C) could be observed with rIL-35 treatment, signifying restrained Th1 but promotion of Th2 and Tregs. *$P < 0.05$, **$P < 0.01$ , ***$P < 0.001$ vs. control: anti-CD3/CD28 (+) rIL-35 (−).

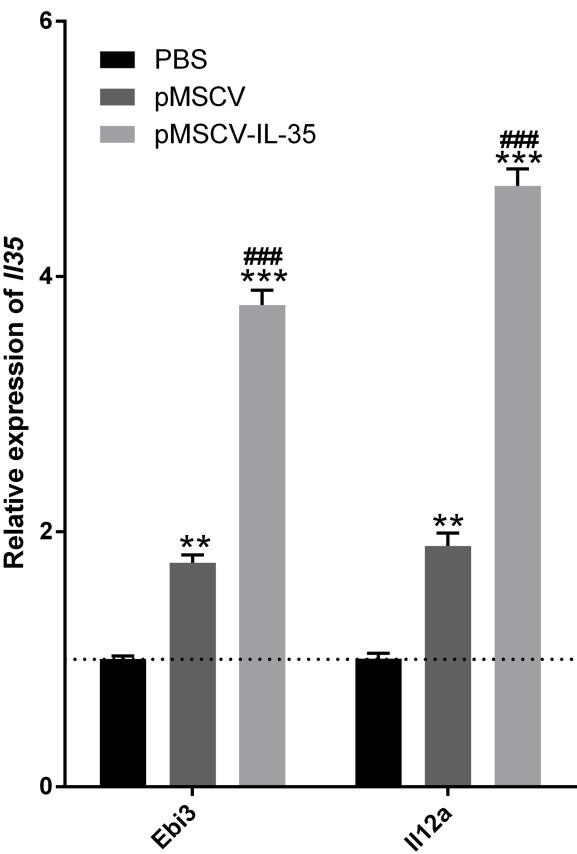

**Figure 4** **The transcriptional levels of *Ebi3* and *Il12a* 72 h post plasmid pMSCV-IL-35-GFP injection.** Both Ebi3 and Il12a, the subunits of IL-35, got nearly doubled in mice exposed to IL-35-carrying plasmid, indicating a successful expression of exogenous IL-35. **$P < 0.01$ , ***$P < 0.001$ vs. PBS group, ###$P < 0.001$ vs. pMSCV group.

but considerably amplified CD4$^+$CD25$^+$ Tregs (Figs. 5E, 5F) compared to PBS group. Consistent with previous reports, Treg cells proliferated in response to IL-35 upregulation (*Castellani et al., 2010*).

## Upregulation of IL-35 increased Th2 cytokine production *in vivo*

In this study, we also measured the mRNA levels of cytokines and genes related to Th1, Th2, CD8$^+$ and Tregs using real time PCR. The results showed that Th1 cytokines IL-2 and IFN-$\gamma$ (Figs. 6A, 6B) as well as Gzmb and Prf1 in CD8$^+$ T cells (Figs. 6D, 6E) obviously increased in mice received pMSCV plasmid. While, the generous expression of IL-35 effectively inhibited the aforementioned effector molecules. Besides, the splenic levels of *Il10* (Fig. 6C) and *Foxp3* (Fig. 6F) of pMSCV-IL-35 group both got remarkably elevated. Therefore, IL-35 could effectively suppress Th1 and CD8$^+$ T cell function, but strengthen Th2 and Tregs.

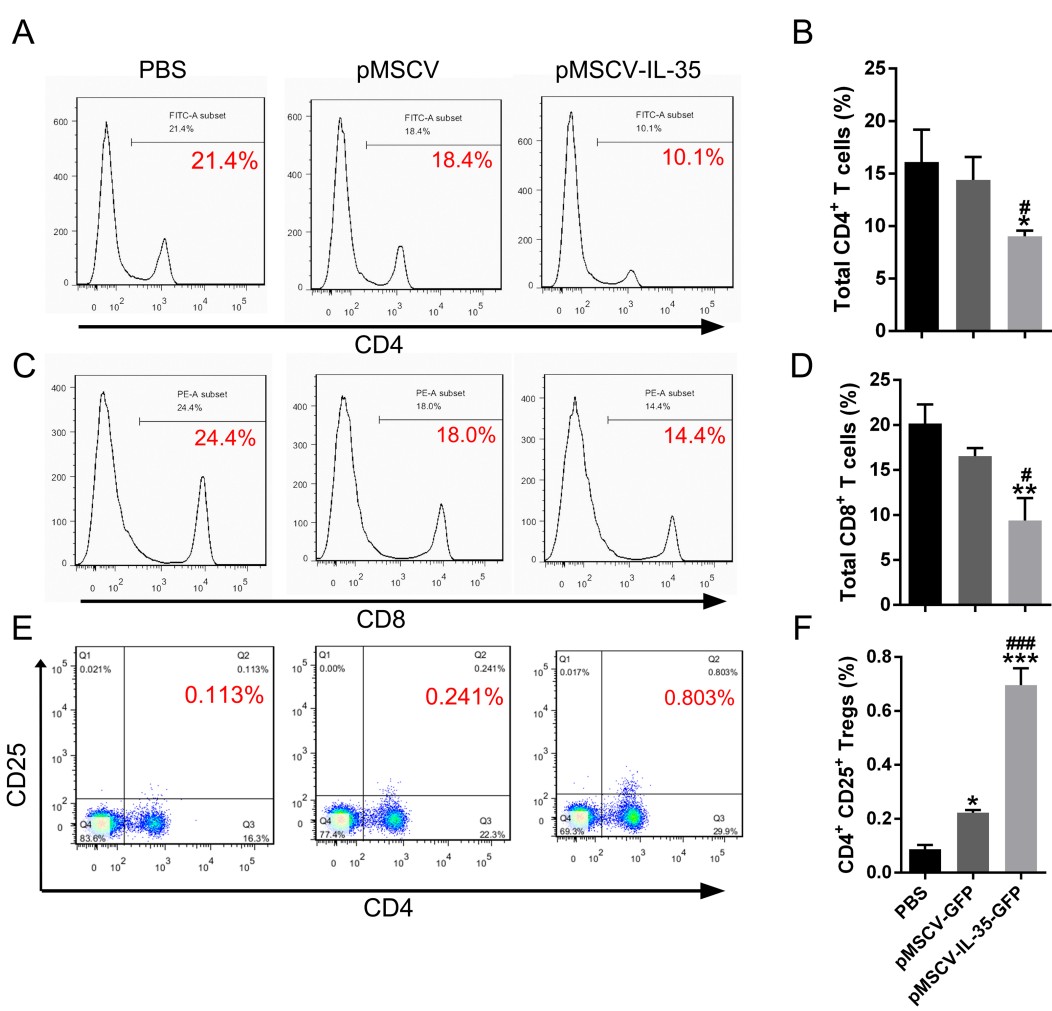

**Figure 5** **The CD4$^+$, CD8$^+$ T cells and CD4$^+$ CD25$^+$ Tregs subsets.** (A), (C) and (E) exhibited FACS detection and (B), (D) and (F) showed the statistical data. IL-35 overexpression *in vivo* clearly inhibited both CD4$^+$ and CD8$^+$ T proliferating but significantly enhanced CD4$^+$ CD25$^+$ Tregs. *$P < 0.05$, **$P$ vs. PBS $< 0.01$, ***$P < 0.001$ vs. PBS group, #$P < 0.05$, ###$P < 0.001$ vs. pMSCV group.

## DISCUSSION

In the present investigation, a plasmid loading IL-35 linear gene containing Ebi3 joint with Il12a was intravenously injected into normal mice to determine the effect of IL-35 heterodimer overexpression on T cell immunity and T cell differentiation under normal condition. Ebi3 and IL-12A, two subunits of IL-35, could be identified 72 h post DNA injection. IL-35 upregulation *in vivo* effectively inhibited CD4$^+$ and CD8$^+$ T cells proliferation and Th1 cytokine secretion. On the contrary, elevated level of IL-35 significantly induced CD4$^+$ CD25$^+$ Tregs and Th2 enhancement. The *in vitro* study provided similar results, suggesting the immunosuppressive action of IL-35 on T-cell subsets evolution. All of the data indicated the Th1 and CD8$^+$ T cells suppression but Th2 and Tregs bias in the presence of IL-35.

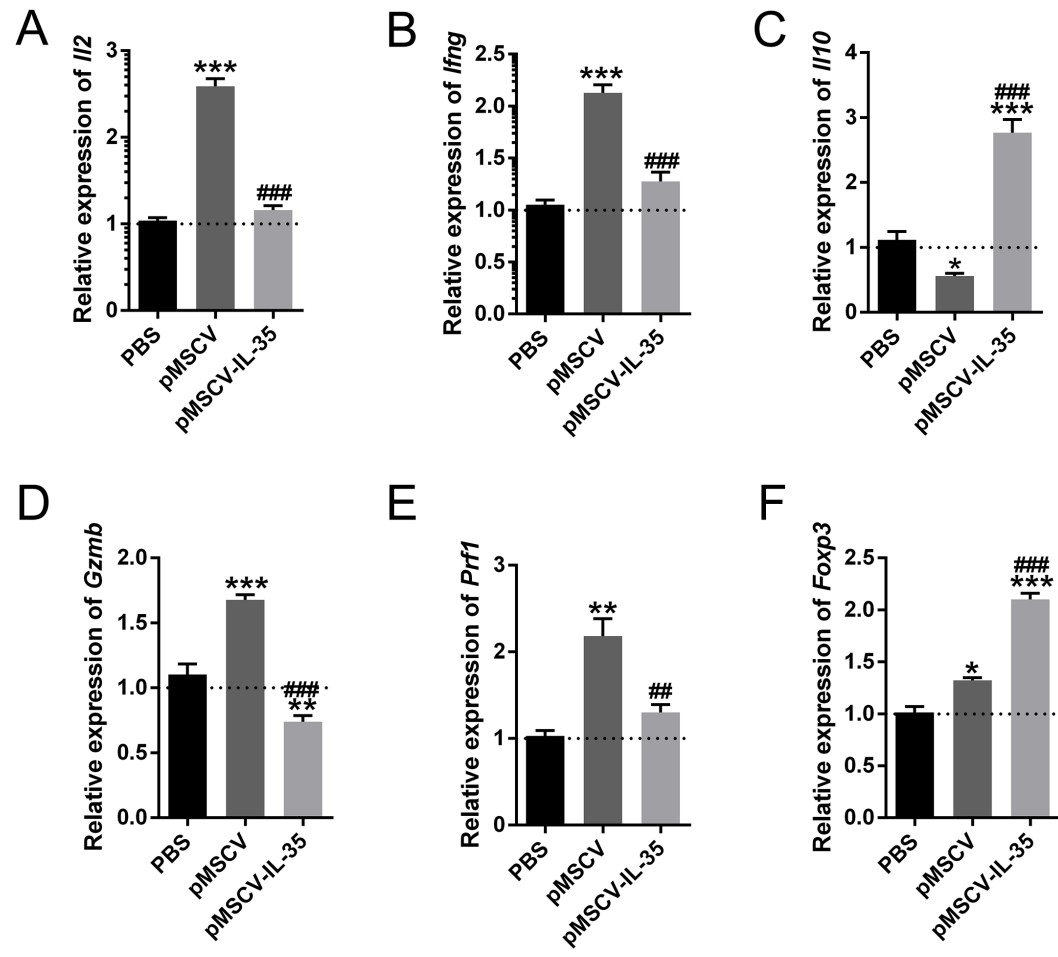

**Figure 6** **The mRNA level of *Il2, Ifng, Il10, Gzmb, Prf1*, and *Foxp3 in vivo*.** Th1 cytokines including (A) *Il2* and (B) *Ifng* together with (D) *Gzmb* and (E) *Prf1* in CD8[+] T cells all got remarkably inhibited, while *Il10* (C) produced by Th2 cells and *Foxp3* (F) in CD4[+] CD25[+] Tregs received cleared upregulation. *$P < 0.05$, **$P < 0.01$, ***$P < 0.001$ vs. PBS group, ##$P < 0.01$, ###$P < 0.001$ vs. pMSCV group.

As we all know, T cells, roughly divided into CD4[+] Th cells, CD8[+] cytotoxic T cells (CTLs) and Tregs according to the functional distinction, play a central role in cell-mediated immunity (*Milstein et al., 2011*; *Overgaard et al., 2015*). Cytokine profiles secreted by these cells further augment their immunoregulatory capacity and act on other subsets (*Biron & Tarrio, 2015*). In brief, IL-2 and IFN-$\gamma$ produced by Th1 cells can enhance the cytotoxic effect of CD8[+] T cells, which can be well inhibited by IL-10 released from Th2 cells and/or IL-35 synthesized by Tregs (*Guo et al., 2017*; *Biron & Tarrio, 2015*). The ability of Th1 and Th2 cytokines has been clearly stated, and recent studies focused on the newly discovered immunosuppressive factor IL-35.

IL-35, a member of the IL-12 family, is a heterodimeric cytokine composed of the Ebi3 and IL-12A assembled in innate Tregs (*Collison et al., 2007*; *Niedbala et al., 2007*). The anti-inflammatory capacity of IL-35 has been investigated in several inflammatory disorder models, such as infection diseases, organ transplantation rejection and autoimmunity

diseases (*Guan et al., 2017*; *Sun et al., 2015*; *Liu et al., 2015*; *Li et al., 2012*). Evidence from these animal models showed that IL-35 administration suppressed inflammation related T cell differentiation. However, the existing results are mainly obtained from disease models, it still remains to be determined whether IL-35 could affect the T subsets evolution without antigen stimulation. The present study primarily aimed at expounding the impact of IL-35 on T cell differentiation and function in normal healthy mice.

Some works proposed that the Tregs and Th2 augment partly at least contributed to the therapeutic potential of IL-35 (*Zhao et al., 2017*). To confirm this finding, activated T cells were treated with recombinant IL-35 factor for 24 h *ex vivo*. The data indicated that IL-35 remarkably suppressed $CD4^+$ T cell proliferation (Fig. 2A) and IL-2 release (Fig. 3A). Instead, $CD4^+CD25^+$ Tregs (Fig. 2B) and IL-10 secretion (Fig. 3B) by Th2 cells got effective increase. Foxp3 also showed a distinct growth under treatment with 50ng/mL IL-35 factor (Fig. 3C). The *in vivo* effect of IL-35 on T-cell subsets development was then investigated after a plasmid carrying IL-35 fragment (Fig. 1) administration in normal mice. The much higher expression of Ebi3 and IL-12p35 in mice received pMSCV-IL-35 plasmid could be observed three days later (Fig. 4). Similarly, both splenic $CD4^+$ and $CD8^+$ T cells performed significantly restrained ( Figs. 5A–5D), but $CD4^+CD25^+$ Tregs got well amplification (Figs. 5E, 5F). Related cytokine and effector molecule analysis further confirmed lower IL-2 and IFN-$\gamma$ level (Figs. 6A, 6B) together with largely reduced Gzmb and Prf1 production (Figs. 6D, 6E) upon IL-35 expression. In contrast, both IL-10 (Fig. 6C) and Foxp3 (Fig. 6F) increased significantly. Thus, IL-35 overexpression indeed affect T cell differentiation and induced an obvious Th2 and Tregs bias.

The suppressive capacity of IL-35 has been tested in many disease models, still the influence of general IL-35 administration on normal body needs to be clarified. Data listed here based on normal healthy mice could partly help to understand how IL-35 work and the potential therapeutic mechanisms in disease cases. And it is important to note that the present study was conducted in normal animals and the outcome might have some difference from those in pathologic condition. The cure mechanisms of IL-35 treatment in various disease models still need further researches.

## CONCLUSIONS

The present study explored the effect of recombinant heterodimer IL-35 on T-cell subset developing in normal mice for the first time. The results indicated that IL-35 upregulation could effectively induce Th2 and Tregs bias and inhibit Th1 and $CD8^+$ T cell function in healthy animals. Meanwhile, further studies concerning long-term effect and related mechanisms are still required to elucidate in IL-35 application.

### Funding
This work was supported by the grant from National Key R&D Program of China (No. 2017YFC1001904), Natural Science Foundation of China for Young Scholars (No.

81501386), and Tianjin Medical University 13th 5-year postgraduate Innovation Fund (No. YJSCX201705). The funders had no role in study design, data collection and analysis, decision to publish, or preparation of the manuscript.

## Grant Disclosures

The following grant information was disclosed by the authors:
National Key R&D Program of China: 2017YFC1001904.
Natural Science Foundation of China: 81501386.
Tianjin Medical University: YJSCX201705.

## Competing Interests

The authors declare there are no competing interests.

## Author Contributions

- Xiaoning Zhang conceived and designed the experiments, performed the experiments, analyzed the data, prepared figures and/or tables, authored or reviewed drafts of the paper.
- Zhiqiang Zhang performed the experiments, contributed reagents/materials/analysis tools.
- Zhiqiang He, Mingyan Ju, Jiaci Li and Jinghua Yuan performed the experiments.
- Yaqing Jing and Keqiu Li conceived and designed the experiments.
- Yi Liu conceived and designed the experiments, analyzed the data, prepared figures and/or tables, authored or reviewed drafts of the paper, approved the final draft.
- Guang Li conceived and designed the experiments, approved the final draft.

## Animal Ethics

The following information was supplied relating to ethical approvals (i.e., approving body and any reference numbers):

Experimental protocols and animal care methods were approved by the Animal Care and Use Committee of Tianjin Medical University (TMUaMEC 2017012).

## Data Availability

The raw measurements are provided in the Supplemental Files.

## Supplemental Information

Supplemental information for this article can be found online at http://dx.doi.org/10.7717/peerj.5638#supplemental-information.

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
