# Peer review of "Interleukin 35 induced Th2 and Tregs bias under normal conditions in mice"

_PeerJ, doi:10.7717/peerj.5638_

## Round 0.1 · original submission · Major Revisions

Your manuscript has been reviewed by two experts in the field and by the Editor. Although your manuscript was well received it cannot be accepted in its present form. If you decide to revise your manuscript please put particular emphasis on improving methods' section, gel figures and a revision of the discussion and conclusion. Moreover the English language needs to be improved as it may lead to misunderstanding of the results/conclusions. My advice would be to get the help of a native speaker when revising the manuscript.

I look forward to receive your revised manuscript

Reviewer 1 ·

Basic reporting

General overview:
Author was trying to investigate the effects modulating IL-35 levels in vivo and in vitro on T cells response and also the effects on T cells subsets in normal mice.

Inappropriate usage of Wordings and phrases found throughout manuscript.

All figures and tables used are appropriate but I could not find any raw data being provided.

Experimental design

Experimental designs are considered up to date even though description on the methodology used are not very details, therefore making replications may not be possible.

Validity of the findings

All the findings are valid and would contribute to important knowledge on IL-35 research and study, but the conclusion part is a bit speculative in which authors suggested that the results indicated a strong evidence of IL-35 in immune-related disorders treatment. This cannot be so since the study was conducted in normal healthy mice. The outcome would have been different if the study is conducted in a pathologic condition in which the immune response would be more complex as compared to healthy and normal condition.

Additional comments

Comments:
1) Overall, the manuscript was not written in good command of English therefore contributing to the lacking in the soundness of its scientific values, even though the experimental procedures used throughout the study are standard and considered up to date.
2) There are a lot of inappropriate usage of wordings and phrases, and grammatical errors throughout the manuscript. Few examples are as follows:
• Line 11 – “disease modelling animals” …. rather than “animal model of diseases”
• Line 12 – the word illuminated is not the appropriate word. Perhaps “elucidated” is much more suitable.
• Line 27 – “…..CD8+ T cells also got remarkably suppression.”…… perhaps it should read “ …. CD8+ T cells was also remarkably suppressed.”
• Line 104 – “….mice got intravenous injected with PBS,….” ……. Perhaps should be written as “….mice received intravenous injection of PBS,….”
• Line 145-146 – “while, the decreased CD4+CD25+ Tregs following anti-CD3/CD28 treatment received certainly amplified”…… this sentence is quite confusing.
• Line 171 – “…..CD8+ T cells exciting, which got nice reversed in IL-35…..”…. this sentence is also quite confusing.
• Line 209 – the word “pressed” …. Does the author mean “suppressed”?

3) In section 2.3, the lymphocytes used were from the spleen tissues of mice, but recombinant human IL-35 was used to induce the cells. Please provide evidence that human IL-35 does work in mice cells.

4) Section 2.9, ANOVA analysis is normally followed by a single post hoc test to confirm the significance. Any post hoc test used here?

5) Line 232 – “….provided strong evidence of its efficacy in immune-related disorders treatment”….. this statement is a bit inappropriate at this stage since the study was conducted in normal healthy mice and not in any pathologic condition. Efficacy of any treatment can only be assessed based on its ability to succesfully cure or at least reduce certain disease state. Since the study was conducted in normal healthy mice, there is always the possibility that the outcome of the study would be different in pathologic condition in which a complex immune response may occur with the release of many important mediators as compared to basic immune response in healthy animals.

Reviewer 2 ·

Basic reporting

-Excellent background but remove reference in(in revision) as this work is still not published or what?
-The English needs some attention. The manuscript need to be revised by a native speaker.
-Manuscript structure is good but gel figs are bad, other figures and tables are relevant and appropriate.

Experimental design

-The basic structure and conclusions are novel. Actually this topic is important
-Research question well defined, relevant & meaningful.
-investigation performed to a high technical & ethical standard
-Methods described with sufficient details.

Validity of the findings

-Conclusive results
-Data is robust, statistically sound, & controlled

-Conclusion are well stated, linked to original research question & limited to supporting results.

Additional comments

The most important point for the present study in discussion; ‘What is the difference of the present study from the others?’The present discussion section should be rewritten

---

## Round 0.2 · Major Revisions

The reviewers have returned a favourable review of your revised manuscript, however there are still issues that need attention:

The gel present in Figure 6 v1 (Fig 1 in v0) is an obvious composite and is (in spite of the requests) of worse quality than that present in v0. It is unclear if what is shown is real or not. The authors should also state the identity of the post hoc test used.

The Authors have also removed (in v1) the section (which was 3.1 in v0) describing the expression and Western blot. I think that portion of the text should not have been removed, and that the gel should be made anew.

Reviewer 1 ·

Basic reporting

No comment

Experimental design

No comment

Validity of the findings

No comment

Additional comments

No comment

---

## Round 0.3 · accepted · Accept

The authors have responded to the editors' queries and amended the manuscript,

#